# Electromagnetohydrodynamic (EMHD) Flow in a Microchannel with Random Surface Roughness

**DOI:** 10.3390/mi14081617

**Published:** 2023-08-16

**Authors:** Nailin Ma, Yanjun Sun, Yongjun Jian

**Affiliations:** 1School of Mathematical Science, Inner Mongolia University, Hohhot 010021, China; nlmamail@163.com (N.M.); sunyanjun.2006@163.com (Y.S.); 2School of Statistics and Mathematics, Inner Mongolia University of Finance and Economics, Hohhot 010070, China

**Keywords:** electromagnetohydrodynamic (EMHD) flow, random wall roughness, microchannel, spectral density, corrugation function

## Abstract

This study investigates the effect of small random transverse wall roughness on electromagnetohydrodynamic (EMHD) flow is in a microchannel, employing the perturbation method based upon stationary random function theory. An exact solution of a random corrugation function *ξ*, which is a measure of the flow rate deviated from the case without the roughness of two plates, is obtained by integrating the spectral density. After the sinusoidal, triangular, rectangular, and sawtooth functions that satisfy the Dirichlet condition are expanded into the Fourier sine series, the spectral density of the sine function is used to represent the corrugation function. Interestingly, for sinusoidal roughness, the final expression of the corrugation function is in good agreement with our previous work. Results show that no matter the shape of the wall roughness, the flow rate always decreases due to the existence of wall corrugation. Variations of the corrugation function and the flow rate strongly depend on fluid wavenumber *λ* and Hartmann number *Ha*. Finally, the flow resistance is found to become small, and the flow rate increases with roughness that is in phase (*θ* = 0) compared with the one that is out of phase (*θ* = *π*).

## 1. Introduction

Studies on dynamic micropumps based on electromagnetohydrodynamic (EMHD) flow and electroosmosis have received much attention in recent years. Micropumps are largely used in a diverse range of fields, including biological and chemical analysis [1], drug delivery [2], microelectronic cooling [3], DNA hybridization, and space exploration [4], among others. In particular, EMHD micropumps offer many advantages over other types of non-mechanical micropumps, including a simple fabrication process, bidirectional pumping ability, and the usability of medium conducting liquid. The mechanism of EMHD flow is based on the Lorentz force, which is generated by applying an electric current to the conducting liquid across the channel in the presence of a perpendicular magnetic field. EMHD flow in microchannels was first proposed by Jang and Lee [5]. Ko and Dulikravich [6] studied non-reflective boundary conditions for EMHD flow simulations. Chakraborty and Paul [7] established a mathematical model to study the comprehensive influence of electromagnetic hydrodynamics on fluid flow control in parallel flat rectangular microchannels. In the absence of an electric field, Awatif et al. [8] analyzed the effect of magnetic force and moderate Reynolds number on MHD Jeffrey hybrid nanofluid through the peristaltic channel: application of cancer treatment. Qian and Bau [9] investigated the basic theory of EMHD in low conductivity solutions and described the various applications of EMHD. Asterios and Eugen [10] considered the electromagnetohydrodynamic (EMHD) free convection flow of a weakly conductive fluid, such as seawater, in an electromagnetic actuator. Buren et al. [11] employed a perturbation method to investigate the problem of EMHD flow in a microchannel with slightly corrugated walls. Tso and Sundaravadivelu [12] investigated the capillary flow between parallel plates in the presence of an electromagnetic field. Si and Jian [13] studied the EMHD micropump in Jeffrey fluid through two parallel corrugated wall microchannels. Wasim et al. [14] studied the dynamic irreversible process of the flow of magnetohybrid nanofluids of Poiseuille in microchannels. Faisal et al. [15] analyzed the effect of pressure gradient on MHD flow of a tri-hybrid Newtonian nanofluid in a circular channel. Bhatti et al. [16] studied simultaneous EMHD dissipative natural convection in microchannels containing porous media saturated with viscoelastic fluids.

However, the above studies are based on smooth microchannel surfaces. In effect, almost all microchannel surfaces have a certain degree of roughness, either incurred during the fabrication process or because of the adsorption adhesion of other species, such as macromolecules [17]. Thus, it is necessary to consider the flow in a rough microchannel. Additionally, surface roughness can be artificially created to support axial rotation or mixing [18,19]. Thus, many researchers have studied the influence of wall roughness on laminar flow since the 1970s. Wang [20] initially investigated how roughness affected the Stokes flow between corrugated plates. Chu [21] studied the effect of wavy surface roughness on the fluid flow in an annulus under microscopic slip conditions. Tashtoush and Al-Odat [22] analyzed the effect of a magnetic field on heat and fluid flow on a wavy surface with a variable heat flux. Chen and Cho [18] studied the mixing characteristics of two-dimensional electroosmotic flow through cosine wavy microchannels. Chang et al. [23] studied the electroosmotic flow in sinusoidal wall microtubules with the perturbation method. Lei et al. [24] used boundary perturbation analysis to conduct theoretical research of electroosmotic pumping in bumpy microtubules. Su et al. [25] studied electro-osmotic flow through a microchannel with corrugated wavy walls under the Debye–Hückel approximation. Ng and Wang [26] studied the Darcy–Brinkman flow through a corrugated channel. Bujurke and Kudenatti [27] ignored the electric field and analyzed the influence of surface roughness on the extrusion film behavior between two rectangular plates and conductive fluid in the presence of transverse magnetic field. Ashmawy [28] investigated the effect of surface roughness on the rate of flow and mean velocity of two stress fluid flows through a sinusoidal corrugated tube.

In previous theoretical studies, wall roughness is generally represented by sine function or cosine function. However, in actual manufacturing, the roughness of the wall is often random. We hope that we can generalize sine or roughness to a more general case. Phan-Thien [29] first carried out the Stokes flow of incompressible Newtonian fluids between corrugated plates with random roughness, employing the theory of stationary stochastic process. The result was in good agreement with that obtained by Wang [21], when wall roughness reduces to a sinusoidal corrugation. Subsequently, Phan-Thien [30] generalized the above results to channels and pipes with parallel stationary random surface roughness. Faltas et al. [31] extended Phan-Thien’s approach [29] to a sparse porous medium to study the effects of Basset-type slip boundary condition and impermeable walls. The results indicated that the slip parameter is a very important factor for almost perfect slip and for high permeability. They also noted that the type of corrugation has an effect on the flow rate and the pressure gradient. Faltas et al. [32] proposed a boundary perturbation method to study an improved micropolar Brinkman model, using a microwave grained cylindrical tube filled with a porous medium through which micropolar fluid flows. At the same time, a stationary stochastic model was used to simulate the surface roughness of the pipe.

However, to our knowledge, no one has studied the effect of random roughness on EMHD flow. Therefore, the main objective of this paper is to generalize the effect of sinusoidal corrugation on EMHD flow to random wall surfaces by using the perturbation method and some properties of stationary random function. The rest of the paper is organized as follows. In Section 2, the perturbation approach is used to construct and solve the EMHD equations defining surface roughness. In Section 3, we introduce the conditions that should be satisfied from for the definitions of random roughness and concrete roughness, and show four different types of roughness. In Section 4, the problem is studied parametrically and the effects of different parameters on the flow are discussed in detail. Section 5 provides a summary of the full text.

## 2. Materials and Methods

We consider the 2D EMHD flow of the viscous, incompressible, conductive Newtonian liquid confined to two fixed corrugated walls with an average distance of 2*H* parts. A Cartesian coordinate system (*x*, *y*, *z*) is adopted with the origin fixed at the middle of the channel. The flow variables are independent of *y*; therefore, the velocity vector has the form:(1)$$u^{*}=u^{*}(x,z)i+w^{*}(x,z)k$$

In our model, the channel length *L* in the *x*-direction and width *W* in the *y*-direction are assumed to be much larger than the layer thickness, that is *W*, *L* >> 2*H*, and the flow is driven both by the pressure gradient and Lorenz force. The channel is subjected to a uniform magnetic field ***B*^*^** in the *z* direction and a uniform electric field ***E*^*^** in the *y* direction, which induce a Lorenz force ***J*^*^**
*× **B*****^*^** in *x*-direction, as shown in Figure 1. Where ***J*^*^** is the electrical current density and ***J*^*^** = *σ*(***E*^*^** + ***u*^*^** × ***B*^*^**) [11], *σ* is the electrical conductivity, ***u*^*^** is the liquid velocity vector with components *u^*^*, *v^*^*, *w^*^* in the *x*, *y*, *z* directions, respectively. 

The dimensional equations of continuity and momentum can be expressed as: (2)$${𝜵}^{*}\cdot u^{*}=0,$$
(3)$$\rho \frac{\partial u^{*}}{\partial t}+\rho (u^{*}\cdot 𝜵)u^{*}=-{𝜵}^{*}p^{*}+\mu {𝜵}^{2}u^{*}+J^{*}\times B^{*}$$
where *p^*^* is the pressure of the liquid, *μ* represents the dynamic viscosity, *ρ* stands for density, and ${𝜵}^{*}=\frac{\partial }{\partial x}i+\frac{\partial }{\partial y}j+\frac{\partial }{\partial z}k$ is the gradient operator. The boundary conditions satisfy no-slip and no-penetration conditions [11], which means that the layer of fluid in contact with a solid body has the same velocity as the body, and the fluid layer molecules do not penetrate the solid layer, are followed as: (4)$$u^{*}=0\;\mathrm{at}\;z^{*}=z_{u}^{*}\;\mathrm{and}\;z^{*}=z_{l}^{*}$$
the upper and lower wavy walls are described by:(5)$$z_{u}^{*}=H+\varepsilon Hm(x),\;z_{l}^{*}=-H\pm \varepsilon Hn(x).$$

When *m*(*x*) = *n*(*x*), we want to study the effect of phase difference on flow, and the plus (mins) sign denotes a situation in which the corrugations are in (half-period out of) phase. The amplitude of the corrugation *ε* << 1, and will be considered a perturbation parameter for the problems described below. *m*(*x*) and *n*(*x*) are stationary random functions, which allows all shapes of ripples to be generalized, and can be expressed as Fourier–Stieltjes integrals [33] (spectral representation of *m*(*x*) and *n*(*x*)):(6)$$m(x)=\int_{-\infty }^{\infty }e^{i\lambda x}dZ_{1}(\lambda ),\;n(x)=\int_{-\infty }^{\infty }e^{i\lambda x}dZ_{2}(\lambda ),$$
where ***i***^2^ = −1, and *dZ*_1_(*λ*), *dZ*_2_(*λ*) are interval random functions that satisfy [33]
(7a)$$\langle dZ_{1}(\lambda )\rangle =\langle dZ_{2}(\lambda )\rangle =0,$$
(7b)$$\langle dZ_{1}(\lambda_{i})\bar{dZ_{1}(\lambda_{j})}\rangle =\delta_{ij}dF_{1}(\lambda_{i})=\delta_{ij}\Omega_{11}(\lambda_{i}),$$
(7c)$$\langle dZ_{1}(\lambda_{i})\bar{dZ_{2}(\lambda_{j})}\rangle =\delta_{ij}dF_{12}(\lambda_{i})=\delta_{ij}\Omega_{12}(\lambda_{i}),$$
(7d)$$\langle dZ_{2}(\lambda_{i})\bar{dZ_{2}(\lambda_{j})}\rangle =\delta_{ij}dF_{2}(\lambda_{i})=\delta_{ij}\Omega_{22}(\lambda_{i}).$$
where the overbar denotes a complex conjugate; *δ_ij_* is the Kronecker delta; and *F*_1_, *F*_12_, *F*_2_ are real non-decreasing bounded functions that can be obtained from the correlation function of *m*(*x*), *n*(*x*). In the general sense, |*R*_11_(*s*)|, |*R*_12_(*s*)|, |*R*_22_(*s*)| tend to zero fast enough, as *s* tends to infinity and has Fourier transforms [33]:(8)$$R_{11}(s)=\langle m(x)m(x+s)\rangle =\int_{-\infty }^{\infty }e^{i\lambda s}dF_{1}(\lambda )=\int_{-\infty }^{\infty }e^{i\lambda s}\Omega_{11}(\lambda )d\lambda ,$$
(9)$$R_{12}(s)=\langle m(x)n(x+s)\rangle =\int_{-\infty }^{\infty }e^{i\lambda s}dF_{12}(\lambda )=\int_{-\infty }^{\infty }e^{i\lambda s}\Omega_{12}(\lambda )d\lambda ,$$
(10)$$R_{22}(s)=\langle n(x)n(x+s)\rangle =\int_{-\infty }^{\infty }e^{i\lambda s}dF_{2}(\lambda )=\int_{-\infty }^{\infty }e^{i\lambda s}\Omega_{22}(\lambda )d\lambda ,$$
(11)$$\Omega_{jk}(\lambda )=\frac{1}{2\pi }\int_{-\infty }^{\infty }e^{-i\lambda s}R_{jk}(s)ds,$$
where Ω*_jk_* (*j*,*k =* 1, 2) are the spectral densities of *m*(*x*) and *n*(*x*).

The Reynolds number is believed to be low, and the momentum balance equation nonlinear inertia can be ignored. We have introduced the stream function *Ψ*, which is given in terms of the velocity components, as follows:(12)$$u=\frac{\partial \psi^{*}}{\partial z^{*}},\;w=-\frac{\partial \psi^{*}}{\partial x^{*}}$$

Supposing the flow is steady, the governing equations of a 2D EMHD flow can be expressed in the dimensional form as:(13)$$-\frac{\partial p^{*}}{\partial x^{*}}+\mu (\frac{\partial^{2}}{\partial x^{*}{}^{2}}+\frac{\partial^{2}}{\partial z^{*}{}^{2}})\frac{\partial \psi^{*}}{\partial z^{*}}+\sigma ΕΒ-\sigma {Β}^{2}\frac{\partial \psi^{*}}{\partial z^{*}}=0,$$
(14)$$-\frac{\partial p^{*}}{\partial z^{*}}-\mu (\frac{\partial^{2}}{\partial x^{*}{}^{2}}+\frac{\partial^{2}}{\partial z^{*}{}^{2}})\frac{\partial \psi^{*}}{\partial x^{*}}=0.$$

Assuming the mean flow is purely in the *x*-direction, and the flow rate per unit width is known to be *Q*. Thus, utilizing *H*, *Q*/2, 1/*H*, and *μQ*/2*H*^2^ as length, stream function, wave number, and pressure scales, respectively, the non-dimensionalization governing Equations (13) and (14) become:(15)$$-\frac{\partial p}{\partial x}+\mu (\frac{\partial^{2}}{\partial x^{2}}+\frac{\partial^{2}}{\partial z^{2}})\frac{\partial \psi }{\partial z}+HaS-Ha^{2}\frac{\partial \psi }{\partial z}=0,$$
(16)$$-\frac{\partial p}{\partial z}-\mu (\frac{\partial^{2}}{\partial x^{2}}+\frac{\partial^{2}}{\partial z^{2}})\frac{\partial \psi }{\partial x}=0,$$
where $Ha=BH\sqrt{\sigma /\mu }$, $S=\frac{2EH^{2}\sqrt{\sigma /\mu }}{Q}$.

Hartmann number Ha is the ratio of Lorentz force to viscous force, and the non-dimensional parameter *S* measures the strength of the external electric field. The dimensionless no-penetration and no-slip boundary conditions are:(17)$$\psi =1,\;\frac{\partial \psi }{\partial n}=0,\;\mathrm{at}\;z=z_{u}=1+\varepsilon m(x),$$
(18)$$\psi =-1,\;\frac{\partial \psi }{\partial n}=0,\;\mathrm{at}\;z=z_{l}=-1\pm \varepsilon n(x).$$
where ***n*** is the normal direction of the channel walls, the normal derivative $\frac{\partial \psi }{\partial n}=n_{x}\frac{\partial \psi }{\partial x}+n_{z}\frac{\partial \psi }{\partial z}$, and *n_x_* and *n_z_* are the *x*-and *z*-components of the normal unit. In view of the randomness of the channel walls, *Ψ* is defined a stochastic stream function, that is, all dynamical variables now become stochastic processes.

Eliminating pressure gradients from Equations (15) and (16), we find that the stream function satisfies:(19)$${\left(\frac{\partial^{2}}{\partial x^{2}}+\frac{\partial^{2}}{\partial z^{2}}\right)}^{2}\psi -Ha^{2}\frac{\partial^{2}\psi }{\partial z^{2}}=0.$$

If there is no roughness, the flow is along *x*-direction and the stream function would be a function of *z* only. However, the presence of surface roughness induces a functional variation in the *x*-direction as well. Therefore, the stream function *Ψ* can be represented by the slight perturbation expansion of *ε*, which takes the following form:(20)$$\psi =\psi_{0}(z)+\varepsilon \psi_{1}(x,z)+\varepsilon^{2}\psi_{2}(x,z)+O(\varepsilon^{2}).$$

Substituting Equation (20) into Equation (19), we obtain differential equations for each power of *ε*:(21)$$\varepsilon^{0}:\frac{d^{4}\psi_{0}}{dz^{4}}-Ha^{2}\frac{d^{2}\psi_{0}}{dz^{2}}=0,$$
(22)$$\varepsilon^{1}:{\left(\frac{\partial^{2}}{\partial x^{2}}+\frac{\partial^{2}}{\partial z^{2}}\right)}^{2}\psi_{1}-Ha^{2}\frac{\partial^{2}\psi_{1}}{\partial z^{2}}=0,$$
(23)$$\varepsilon^{2}:{\left(\frac{\partial^{2}}{\partial x^{2}}+\frac{\partial^{2}}{\partial z^{2}}\right)}^{2}\psi_{2}-Ha^{2}\frac{\partial^{2}\psi_{2}}{\partial z^{2}}=0.$$

The boundary conditions Equations (17) and (18) can be expanded in Taylor series about the mean wall positions *z* = 1, and *z* = −1. From which the boundary conditions become:(24)$$\psi_{0}(1)+\varepsilon \{\psi_{1}(1)+m(x)\frac{\partial \psi_{0}}{\partial z}(1)\}+\varepsilon^{2}\{\psi_{2}(1)+m(x)\frac{\partial \psi_{1}}{\partial z}(1)+\frac{1}{2}m^{2}(x)\frac{\partial^{2}\psi_{0}}{\partial z^{2}}(1)\}+O(\varepsilon^{2})=1.$$
(25)$$\psi_{0}(-1)+\varepsilon \{\psi_{1}(-1)\pm n(x)\frac{\partial \psi_{0}}{\partial z}(-1)\}+\varepsilon^{2}\{\psi_{2}(-1)\pm n(x)\frac{\partial \psi_{1}}{\partial z}(-1)+\frac{1}{2}n^{2}(x)\frac{\partial^{2}\psi_{0}}{\partial z^{2}}(-1)\}+O(\varepsilon^{2})=-1.$$
(26)$$\begin{array}{c} \frac{\partial \psi }{\partial n}(1)=\frac{\nabla \psi \cdot \nabla (z-(1+\varepsilon m(x)))}{\left|z-(1+\varepsilon m(x))\right|}\\ =\frac{\partial \psi_{0}}{\partial z}(1)+\varepsilon \{-m^{\prime }(x)\frac{\partial \psi_{0}}{\partial x}(1)+\frac{\partial \psi_{1}}{\partial z}(1)+m(x)\frac{\partial^{2}\psi_{0}}{\partial z^{2}}(1)\}+\varepsilon^{2}\{-m^{\prime }(x)\frac{\partial \psi_{1}}{\partial x}(1)-m(x)m^{\prime }(x)\frac{\partial^{2}\psi_{0}}{\partial x\partial z}(1)+\frac{\partial \psi_{2}}{\partial z}(1)+\\ m(x)\frac{\partial^{2}\psi_{1}}{\partial z^{2}}(1)+\frac{1}{2}m^{2}(x)\frac{\partial^{3}\psi_{0}}{\partial z^{3}}(1)+O(\varepsilon^{2})=0. \end{array}$$
(27)$$\begin{array}{c} \frac{\partial \psi }{\partial n}(-1)=\frac{\nabla \psi \cdot \nabla (z-(-1\pm \varepsilon n(x)))}{\left|z-(-1\pm \varepsilon n(x))\right|}\\ =\frac{\partial \psi_{0}}{\partial z}(-1)+\varepsilon \{\mp n^{\prime }(x)\frac{\partial \psi_{0}}{\partial x}(-1)+\frac{\partial \psi_{1}}{\partial z}(-1)\pm n(x)\frac{\partial^{2}\psi_{0}}{\partial z^{2}}(-1)\}+\varepsilon^{2}\{\mp n^{\prime }(x)\frac{\partial \psi_{1}}{\partial x}(-1)-n(x)n^{\prime }(x)\frac{\partial^{2}\psi_{0}}{\partial x\partial z}(-1)\\ +\frac{\partial \psi_{2}}{\partial z}(-1)\pm n(x)\frac{\partial^{2}\psi_{1}}{\partial z^{2}}(-1)+\frac{1}{2}n^{2}(x)\frac{\partial^{3}\psi_{0}}{\partial z^{3}}(-1)-\frac{1}{2}{\left[n^{\prime }(x)\right]}^{2}\frac{\partial \psi_{0}}{\partial z}(-1)\}+O(\varepsilon^{2})=0. \end{array}$$

In boundary conditions, the argument “*±*1” denotes the evaluations at *z* = ±1, and the prime denotes a derivative with respect to the argument.

The boundary conditions of leading order problem are:(28)$$\varepsilon^{0}:\;\psi_{0}(1)=1,\;\psi_{0}(-1)=-1,\;\frac{\partial \psi_{0}}{\partial z}(1)=0,\;\frac{\partial \psi_{0}}{\partial z}(-1)=0.$$

The leading order *ε*^0^-solution yields:(29)$$\psi_{0}=A_{1}[z-\frac{\sinh(Haz)}{Ha\cosh(Ha)}],$$
where $A_{1}=\frac{Ha}{Ha-\tanh(Ha)}$.

The boundary conditions of first-order problem are:$$\varepsilon^{1}:\;\psi_{1}(1)=0,\;\psi_{1}(-1)=0,$$
(30)$$\frac{\partial \psi_{1}}{\partial z}(1)=\frac{m(x)Ha^{2}}{Ha\coth(Ha)-1},\;\frac{\partial \psi_{1}}{\partial z}(-1)=\pm \frac{n(x)Ha^{2}}{1-Ha\coth Ha}.$$

Based on the expressions of boundary conditions, assuming that the *ε*^1^-solution is a stationary random function with the following spectral representation:(31)$$\psi_{1}=\int_{-\infty }^{\infty }\Phi_{1}(y;\lambda )e^{i\lambda x}dZ_{1}(\lambda )+\int_{-\infty }^{\infty }\Phi_{2}(y;\lambda )e^{i\lambda x}dZ_{2}(\lambda )$$
where Φ***_j_*** (*y; λ*) (*j =* 1,2) are functions of *y* parametrized by *λ*. Both Φ_1_ and Φ_2_ hold:(32)$$\frac{d^{4}\Phi_{j}}{dz^{4}}-(Ha^{2}+2\lambda^{2})\frac{d^{2}\Phi_{j}}{dz^{2}}+\lambda^{4}=0,(j=1,2)$$
where
(33)$$\Phi_{1}(1)=0,\;\Phi_{1}(-1)=0,\;\Phi_{1}{}^{\prime }(1)=\frac{Ha^{2}}{Ha\coth(Ha)-1},\;\Phi_{1}{}^{\prime }(-1)=0,$$
(34)$$\Phi_{2}(1)=0,\;\Phi_{2}(-1)=0,\;\Phi_{2}{}^{\prime }(1)=0,\;\Phi_{2}{}^{\prime }(-1)=\pm \frac{Ha^{2}}{1-Ha\coth(Ha)}.$$

The solutions are:(35)$$\Phi_{j}(z,\lambda )=\alpha_{j}e^{k_{1}z}+\beta_{j}e^{-k_{1}z}+\gamma_{j}e^{k_{2}z}+\delta_{j}e^{-k_{2}z}.$$
where *α_j_*, *β_j_*, *γ_j_*, *δ_j_* (*j =* 1,2) are constants and given by:(36)$$\alpha_{1}=\pm \beta_{2}=e^{-k_{1}-2k_{2}}Ha^{2}(e^{2k_{1}+4k_{2}}(k_{1}-k_{2})+2k_{2}e^{2k_{2}}-e^{2k_{1}}(k_{1}+k_{2}))/D,$$
(37)$$\beta_{1}=\pm \alpha_{2}=e^{-k_{1}-2k_{2}}Ha^{2}((-1+e^{4k_{2}})k_{1}+(1+e^{4k_{2}}-2e^{2(k_{1}+k_{2})})k_{2})/D,$$
(38)$$\delta_{1}=\pm \gamma_{2}=e^{-2k_{1}-k_{2}}Ha^{2}(2k_{1}e^{2k_{1}}-e^{2k_{2}}(k_{1}+e^{4k_{1}}(k_{1}-k_{2})+k_{2}))/D,$$
(39)$$\gamma_{1}=\pm \delta_{2}=e^{-2k_{1}-k_{2}}Ha^{2}((1+e^{4k_{1}}-2e^{2(k_{1}+k_{2})})k_{1}+k_{2}(-1+e^{4k_{1}}))/D,$$
where,
(40a)$$D=4(-1+Ha\coth(Ha)(2k_{1}k_{2}-2k_{1}k_{2}\cosh(2k_{1})\cosh(2k_{2})+\sinh(2k_{1})\sinh(2k_{2})(k_{1}{}^{2}+k_{2}{}^{2}))$$
(40b)$$k_{1}=\sqrt{\frac{2\lambda^{2}+Ha^{2}+Ha\sqrt{Ha^{2}+4\lambda^{2}}}{2}},$$
(40c)$$k_{2}=\sqrt{\frac{2\lambda^{2}+Ha^{2}-Ha\sqrt{Ha^{2}+4\lambda^{2}}}{2}}$$

The mean value of *Ψ*_1_ is zero, due to Equations (7a) and (31). As a result, the mean of stream function depends on the term of order *O*(*ε^2^*). The boundary conditions of second-order problem are:(41)$$\varepsilon^{2}:\;\psi_{2}(1)=\frac{-m^{2}(x)Ha^{2}}{2(Ha\coth Ha-1)},\;\psi_{2}(-1)=\frac{n^{2}(x)Ha^{2}}{2(Ha\coth Ha-1)},$$
$$\frac{\partial \psi_{2}}{\partial z}(1)=[\frac{Ha^{3}}{2(Ha-\tanh(Ha))}-{\Phi_{1}}^{\prime \prime }(1,\lambda )-{\Phi_{2}}^{\prime \prime }(1,\lambda )]m^{2}(x),$$
(42)$$\frac{\partial \psi_{2}}{\partial z}(-1)=[\frac{Ha^{3}}{2(Ha-\tanh(Ha))}\mp {\Phi_{1}}^{\prime \prime }(-1,\lambda )\mp {\Phi_{2}}^{\prime \prime }(-1,\lambda )]n^{2}(x).$$

Similar to the process of *ε*^1^-solution representation, the solution to second-order problem can be written as:(43)$$\begin{array}{c} \psi_{2}(x,z)=\int_{-\infty }^{\infty }\int_{-\infty }^{\infty }\zeta_{11}(z,\lambda ,\lambda \prime )]e^{i(\lambda -\lambda \prime )x}dZ_{1}(\lambda )\bar{dZ_{1}(\lambda \prime )}+\int_{-\infty }^{\infty }\int_{-\infty }^{\infty }\zeta_{12}(z,\lambda ,\lambda \prime )]e^{i(\lambda -\lambda \prime )x}dZ_{1}(\lambda )\bar{dZ_{2}(\lambda \prime )}\\ \;\;\;+\int_{-\infty }^{\infty }\int_{-\infty }^{\infty }\zeta_{22}(z,\lambda ,\lambda \prime )]e^{i(\lambda -\lambda \prime )x}dZ_{2}(\lambda )\bar{dZ_{2}(\lambda \prime )}. \end{array}$$
where *ζ_jk_* (*z*, *λ*, *λ*′) are functions of *z* parameterized by *λ* and *λ*′. All three functions *ζ_jk_* (*j*, *k =* 1, 2) satisfy:(44)$$\frac{d^{4}\zeta_{j}{}_{k}}{dz^{4}}-(Ha^{2}+2{\left(\lambda -\lambda^{\prime }\right)}^{2})\frac{d^{2}\zeta_{jk}}{dz^{2}}+{\left(\lambda -\lambda^{\prime }\right)}^{4}=0,$$
with the following boundary conditions:$$\zeta_{11}(1)=\frac{-Ha^{2}}{2\left(Ha\coth Ha-1\right)},\;\zeta_{12}(1)=0,\;\zeta_{22}(1)=0,$$
$$\zeta_{11}(-1)=\frac{Ha^{2}}{2\left(Ha\coth Ha-1\right)},\;\zeta_{12}(-1)=0,\;\zeta_{22}(-1)=0,$$
$$\zeta_{11}{}^{\prime }(1)=\frac{Ha^{3}}{2\left(Ha-\tanh Ha\right)}-{\Phi_{1}}^{\prime \prime }(1,\lambda ),\;\zeta_{12}{}^{\prime }(1)=-{\Phi_{2}}^{\prime \prime }(1,\lambda ),\;\zeta_{22}{}^{\prime }(1)=0,$$
(45)$$\zeta_{11}{}^{\prime }(-1)=0,\;\zeta_{12}{}^{\prime }(-1)=\mp {\Phi_{1}}^{\prime \prime }\left(-1,\lambda \right),\;\zeta_{22}{}^{\prime }(-1)=\frac{Ha^{3}}{2\left(Ha-\tanh Ha\right)}\mp {\Phi_{2}}^{\prime \prime }(-1,\lambda ).$$

Note also that:(46)$${\Phi_{1}}^{\prime \prime }(1,\lambda )=k_{1}{}^{2}(\alpha_{1}e^{k_{1}}+\beta_{1}e^{-k_{1}})+k_{2}{}^{2}(\gamma_{1}e^{k_{2}}+\delta_{1}e^{-k_{2}}),$$
(47)$${\Phi_{1}}^{\prime \prime }(-1,\lambda )=k_{1}{}^{2}(\alpha_{1}e^{-k_{1}}+\beta_{1}e^{k_{1}})+k_{2}{}^{2}(\gamma_{1}e^{-k_{2}}+\delta_{1}e^{k_{2}}),$$
(48)$${\Phi_{2}}^{\prime \prime }(1,\lambda )=k_{1}{}^{2}(\alpha_{2}e^{k_{1}}+\beta_{2}e^{-k_{1}})+k_{2}{}^{2}(\gamma_{2}e^{k_{2}}+\delta_{2}e^{-k_{2}})=\pm [k_{1}{}^{2}(\alpha_{1}e^{-k_{1}}+\beta_{1}e^{k_{1}})+k_{2}{}^{2}(\gamma_{1}e^{-k_{2}}+\delta_{1}e^{k_{2}})],$$
(49)$${\Phi_{2}}^{\prime \prime }(-1,\lambda )=k_{1}{}^{2}(\alpha_{2}e^{-k_{1}}+\beta_{2}e^{k_{1}})+k_{2}{}^{2}(\gamma_{2}e^{-k_{2}}+\delta_{2}e^{k_{2}})=\pm [k_{1}{}^{2}(\alpha_{1}e^{k_{1}}+\beta_{1}e^{-k_{1}})+k_{2}{}^{2}(\gamma_{1}e^{k_{2}}+\delta_{1}e^{-k_{2}})].$$

Although the detailed *ε*^2^-solution can be calculated, we do not need to know its exact result if one is only interested in the mean quantity. According to Equations (7b–d) and (43), we have:(50)$$\langle \psi_{2}\rangle =\sum_{j,k=1,2}\int_{-\infty }^{\infty }\zeta_{jk}(z;\lambda ,\lambda^{\prime })\Omega_{jk}(\lambda )d\lambda .$$

Since the mean of the first-order solutions of the steam function is already zero, in order for the mean of the second-order solutions not being zero, this requires:(51)$$\lambda =\lambda^{\prime }.$$

Therefore, the second-order problem of Equation (44) turns into:(52)$$\frac{d^{4}\zeta_{jk}}{dz^{4}}-Ha^{2}\frac{d^{2}\zeta_{jk}}{dz^{2}}=0.$$

The fourth-order ordinary differential equation has the following general solution:(53)$$\zeta_{jk}(z)=a_{jk}+b_{jk}z+c_{jk}e^{Haz}+d_{jk}e^{-Haz},$$
where *a_jk_*, *b_jk_*, *c_jk_*, *d_jk_* can be determined by boundary conditions. Only the coefficients *b_jk_* are needed, and the other coefficients have no contribution to our present problem. The expressions of *b_jk_* can be given as:(54)$$b_{11}=-\frac{\sinh Ha(((-2Ha{\Phi_{1}}^{\prime \prime }(1,\lambda )+3Ha^{3})\cosh Ha+2{\Phi_{1}}^{\prime \prime }\left(1,\lambda )\sinh Ha\right)}{4{\left(-Ha\cosh Ha+\sinh Ha\right)}^{2}}$$
(55)$$b_{12}=\frac{\sinh Ha(\pm {\Phi_{1}}^{\prime \prime }(-1,\lambda )+{\Phi_{2}}^{\prime \prime }(1,\lambda ))}{2Ha\cosh Ha-2\sinh Ha},$$
(56)$$b_{22}=\frac{\sinh(Ha)[\pm 2{\Phi_{2}}^{\prime \prime }(-1,\lambda )(Ha\cosh Ha-\sinh Ha)-Ha^{3}\cosh Ha]}{4{\left(-Ha\cosh Ha+\sinh Ha\right)}^{2}}.$$

Substituting Equation (20) into Equation (15), it can be obtained as:(57)$$\begin{array}{c} -\frac{\partial p}{\partial x}+HaS=-(\frac{\partial^{2}}{\partial x^{2}}+\frac{\partial^{2}}{\partial z^{2}})\frac{\partial \psi }{\partial z}+Ha^{2}\frac{\partial \psi }{\partial z}\\ =(-\frac{\partial^{3}\psi_{0}}{\partial z\partial x^{2}}-\frac{\partial^{3}\psi_{0}}{\partial z^{3}}+Ha^{2}\frac{\partial \psi_{0}}{\partial z})-\varepsilon (\frac{\partial^{3}\psi_{1}}{\partial z\partial x^{2}}+\frac{\partial^{3}\psi_{1}}{\partial z^{3}}-Ha^{2}\frac{\partial \psi_{1}}{\partial z})-\varepsilon^{2}(\frac{\partial^{3}\psi_{2}}{\partial z\partial x^{2}}+\frac{\partial^{3}\psi_{2}}{\partial z^{3}}-Ha^{2}\frac{\partial \psi_{2}}{\partial z}). \end{array}$$

Averaging Equation (57) over one wavelength of corrugations, and inserting the relevant results of zero-order and second-order solutions, we have:(58)$$\begin{array}{c} -\langle \frac{\partial p}{\partial x}\rangle +HaS=Ha^{2}A_{1}-\varepsilon^{2}\langle \frac{\partial^{3}\psi_{2}}{\partial z\partial x^{2}}+\frac{\partial^{3}\psi_{2}}{\partial z^{3}}-Ha^{2}\frac{\partial \psi_{2}}{\partial z}\rangle \\ \;\;\;\;\;\;\;\;\;\;\;=A_{1}Ha^{2}(1+\varepsilon^{2}\int_{-\infty }^{\infty }\frac{\sum b_{jk}\Omega_{jk}(\lambda )}{A_{1}}d\lambda )=A_{1}Ha^{2}(1+\varepsilon^{2}\xi ), \end{array}$$
where:(59a)$$\xi =\int_{-\infty }^{\infty }\frac{\sum_{j,k=1,2}b_{jk}\Omega_{jk}(\lambda )}{A_{1}}d\lambda ,$$
(59b)$$\langle \frac{\partial p}{\partial x}\rangle =\frac{1}{\lambda }\int_{0}^{\lambda }\frac{\partial p}{\partial x}dx$$

From now, *ξ* is called as corrugation function used to describe the effect of the random roughness on EMHD flow. It can be seen that from Equation (16) over one wavelength of corrugations, the mean pressure gradient in *z*-direction is zero. Applying the above dimensionless parameters associated with pressure gradient and the definition of *S*, Equation (58) can be inverted into:(60)$$\frac{Q}{2H^{3}(-\langle \frac{\partial p}{\partial x}\rangle +\sigma \mathrm{EB})/(A_{1}Ha^{2}\mu )}=1-\varepsilon^{2}\xi +O(\varepsilon^{2}).$$

The flow rate increases with the corrugations when *ξ* is negative and decreases with the corrugations when *ξ* is positive for a certain mean pressure drop and electromagnetic force. When the mean pressure gradient is taken as zero, the flow becomes a pure EMHD flow.

The coefficient of *ξ* depends on the parameters *Ha* and *λ*, which show the impact of corrugations on the driving power required to drive the particular flow rate Q (per unit width of the channel) of flow perpendicular to the corrugations. The driving force is the sum of the mean pressure drop and the electromagnetic force σ*EB*.

## 3. Dirichlet’s Condition and Applications

It is assumed that *m*(*x*) and *n*(*x*) satisfy Dirichlet’s condition, using harmonic properties of the corrugation function, they can be represented as the Fourier sine series:(61)$$m(x)=\sum_{n=1}^{\infty }a_{n}\sin n\lambda x,\;n(x)=\sum_{n=1}^{\infty }b_{n}\sin n\lambda x,$$
where *λ* is wave number, *a_n_* and *b_n_* are amplitudes, and index n denotes a natural number in the followings for all kinds of corrugations. Under this situation, spectral densities can be expressed as:(62a)$$\Omega_{11}(\lambda )=\frac{1}{4}\sum_{n=1}^{\infty }a_{n}{}^{2}(\delta (\omega -n\lambda )+\delta (\omega +n\lambda )),$$
(62b)$$\Omega_{12}(\lambda )=\frac{1}{4}\sum_{n=1}^{\infty }a_{n}b_{n}(\delta (\omega -n\lambda )+\delta (\omega +n\lambda )),$$
(62c)$$\Omega_{22}(\lambda )=\frac{1}{4}\sum_{n=1}^{\infty }b_{n}{}^{2}(\delta (\omega -n\lambda )+\delta (\omega +n\lambda )),$$
where *δ* is the Dirac generalized function. Any shape of the corrugation represented as the Fourier series can be solved in this way. Previous studies have shown that the phase difference between two corrugated walls has significant effects on the flow. This makes us consider two cases where the upper and lower corrugated walls are either in phase or half-period out of phase. Here, ’in phase’ corresponds to the exact same direction of the corrugations on the upper and lower walls, while ‘half-period out of phase’ means the opposite direction of the corrugations on the higher and lower walls. As particular examples of the above study, we consider four types of corrugations: sinusoidal, triangular, rectangular, and sawtooth corrugations.

### 3.1. Sinusoidal Corrugations

Under the sinusoidal corrugations, *m*(*x*) and *n*(*x*) can be represented as follows:(63)m(x)=sin(λx), n(x)=sin(λx+θ).
with *n* = 1 and *a_n_* = *b_n_* = 1 in Equation (61). In this case, the spectral densities and corresponding corrugation function can be written as:(64a)$$\Omega_{11}(\lambda )=\Omega_{22}(\lambda )=\frac{1}{4}[\delta (\omega -\lambda )+\delta (\omega +\lambda )],$$
(64b)$$\Omega_{12}(\lambda )=\frac{1}{4}\cos\theta [\delta (\omega -\lambda )+\delta (\omega +\lambda )]+\frac{1}{4i}\sin\theta [\delta (\omega -\lambda )-\delta (\omega +\lambda )].$$
(65)$$\begin{array}{c} \xi_{1}=\int_{-\infty }^{\infty }\frac{b_{jk}(\lambda ,Ha)\Omega_{jk}(\lambda )}{2A_{1}}d\lambda \\ =\{e^{-k_{1}-k_{2}}\sinh Ha[-e^{k_{1}+k_{2}}Ha^{3}\cosh Ha+(Ha\cosh Ha-\sinh Ha)e^{k_{2}}k_{1}^{2}\times \\ (\alpha_{1}\cos\theta +\beta_{1}+e^{2k_{1}}(\alpha_{1}+\beta_{1}\cos\theta ))+e^{k_{1}}k_{2}^{2}(\gamma_{1}\cos\theta +\delta_{1}+e^{2k_{2}}(\gamma_{1}+\delta_{1}\cos\theta ))]\}/[2A_{1}{\left(-Ha\cosh Ha+\sinh Ha\right)}^{2}]. \end{array}$$

Interestingly, by substituting the relevant coefficients *α*_1_, *β*_1_, *γ*_1_, *δ*_1_, *k*_1_, and *k*_2_ in Equations (36)–(40) into Equation (65), we find that the final analytical expression of the corrugation function *ξ*_1_ is the exactly same as the results obtained by Buren and Jian [34].

### 3.2. Triangular Corrugations

The EMHD flow diagram of the triangular corrugations are shown in Figure 2.

For the triangular corrugations, *m*(*x*) and *n*(*x*) can be represented as:(66)$$m(x)=\left\{\begin{array}{l} \frac{2}{\pi }\lambda x,0<x<\frac{\pi }{2\lambda }\\ 2-\frac{2}{\pi }\lambda x,\frac{\pi }{2\lambda }<x<\frac{3\pi }{2\lambda }\\ \frac{2}{\pi }\lambda x-4,\frac{3\pi }{2\lambda }<x<\frac{2\pi }{\lambda } \end{array}\right.,\;n(x)=\left\{\begin{array}{l} \frac{2}{\pi }\lambda x,0<x<\frac{\pi }{2\lambda }\\ 2-\frac{2}{\pi }\lambda x,\frac{\pi }{2\lambda }<x<\frac{3\pi }{2\lambda }\\ \frac{2}{\pi }\lambda x-4,\frac{3\pi }{2\lambda }<x<\frac{2\pi }{\lambda } \end{array}\right..$$

The Fourier coefficient *a_n_* and *b_n_* can be determined as:$$a_{n}=b_{n}=\frac{8{\left(-1\right)}^{n-1}}{\pi^{2}{\left(2n-1\right)}^{2}}.$$

Corresponding spectral densities and the corrugation function *ξ*_2_ are:(67)$$\Omega_{11}(\lambda )=\Omega_{22}(\lambda )=\Omega_{12}(\lambda )=\frac{1}{4}\sum_{n=1}^{\infty }a_{n}^{2}[\delta (\omega -n\lambda )+\delta (\omega +n\lambda )],$$
(68)$$\begin{array}{c} \xi_{2}=\sum_{n=1}^{\infty }a_{n}{}^{2}\frac{b_{jk}(n\lambda ,Ha)}{2A_{1}}=\sum_{n=1}^{\infty }\frac{64b_{jk}(n\lambda ,Ha)}{2\pi^{4}{\left(2n-1\right)}^{4}A_{1}}\\ \;=\sum_{n=1}^{\infty }\frac{-64Ha^{3}k_{2n}\cosh Ha+Ha^{2}\coth k_{2n}((k_{2n}^{2}-k_{1n}^{2})\sinh Ha+Hak_{1n}\cosh Ha\tanh k_{1n})}{2\pi^{4}{\left(2n-1\right)}^{4}A_{1}-1+Ha\coth Ha)(Ha\cosh Ha-\sinh Ha)(k_{2n}-k_{1n}\coth k_{2n}\tanh k_{1n})} \end{array}$$
where,
(69a)$$k_{1n}=\sqrt{\frac{2n^{2}\lambda^{2}+Ha^{2}+Ha\sqrt{Ha^{2}+4n^{2}\lambda^{2}}}{2}},$$
(69b)$$k_{2n}=\sqrt{\frac{2n^{2}\lambda^{2}+Ha^{2}-Ha\sqrt{Ha^{2}+4n^{2}\lambda^{2}}}{2}}.$$

### 3.3. Rectangular Corrugations

The EMHD flow diagram of the rectangular corrugations are shown in Figure 3.

In case of the rectangular corrugations, *m*(*x*) and *n*(*x*) can be represented as:(70)$$m(x)=\left\{\begin{array}{l} 1,0<x<\frac{\pi }{\lambda }\\ -1,\frac{\pi }{\lambda }<x<\frac{2\pi }{\lambda } \end{array}\right.,\;n(x)=\left\{\begin{array}{l} 1,0<x<\frac{\pi }{\lambda }\\ -1,\frac{\pi }{\lambda }<x<\frac{2\pi }{\lambda } \end{array}\right..$$

The equations of the upper and lower walls are then given, respectively:(71)$$z=z_{u}=1+\varepsilon \sum_{n=1}^{\infty }a_{n}\sin(2n-1)\lambda x,\;z=z_{l}=-1\pm \varepsilon \sum_{n=1}^{\infty }a_{n}\sin(2n-1)\lambda x.$$
where $a_{n}=\frac{4}{\pi (2n-1)}$.

Corresponding spectral densities and the corrugation function *ξ*_3_ are:(72)$$\Omega_{11}(\lambda )=\Omega_{22}(\lambda )=\Omega_{12}(\lambda )=\frac{1}{4}\sum_{n=1}^{\infty }a_{n}^{2}[\delta (\omega -n\lambda )+\delta (\omega +n\lambda )],$$
(73)$$\begin{array}{c} \xi_{3}=\sum_{n=1}^{\infty }a_{n}{}^{2}\frac{b_{jk}(n\lambda ,Ha)}{2A_{1}}=\sum_{n=1}^{\infty }\frac{16b_{jk}(n\lambda ,Ha)}{2\pi^{2}{\left(2n-1\right)}^{2}A_{1}}\\ \;=\sum_{n=1}^{\infty }\frac{-16Ha^{3}\cdot k_{2n}\cosh Ha+Ha^{2}\coth k_{2n}\cdot ((k_{2n}^{2}-k_{1n}^{2})\sinh Ha+Ha\cdot k_{1n}\cosh Ha\tanh k_{1n})}{2\pi^{2}{\left(2n-1\right)}^{2}A_{1},(-1+Ha\coth Ha)(Ha\cosh Ha-\sinh Ha)(k_{2n}-k_{1n}\coth k_{2n}\tanh k_{1n})} \end{array}$$
where,
(74a)$$k_{1n}=\sqrt{\frac{2n^{2}\lambda^{2}+Ha^{2}+Ha\sqrt{Ha^{2}+4n^{2}\lambda^{2}}}{2}},$$
(74b)$$k_{2n}=\sqrt{\frac{2n^{2}\lambda^{2}+Ha^{2}-Ha\sqrt{Ha^{2}+4n^{2}\lambda^{2}}}{2}}.$$

### 3.4. Sawtooth Corrugations

The EMHD flow diagram of the sawtooth corrugations are shown in Figure 4.

For the sawtooth corrugations, *m*(*x*) and *n*(*x*) can be represented as:(75)$$m(x)=\left\{\begin{array}{l} \frac{\lambda }{\pi }x,0<x<\frac{\pi }{\lambda }\\ \frac{\lambda }{\pi }x-1,\frac{\pi }{\lambda }<x<\frac{2\pi }{\lambda } \end{array}\right.,\;n(x)=\left\{\begin{array}{l} \frac{\lambda }{\pi }x,0<x<\frac{\pi }{\lambda }\\ \frac{\lambda }{\pi }x-1,\frac{\pi }{\lambda }<x<\frac{2\pi }{\lambda } \end{array}\right..$$

The equations of the upper and lower walls are then given respectively as:(76)$$z=z_{u}=1+\varepsilon \sum_{n=1}^{\infty }a_{n}\sin(2n-1)\lambda x,\;z=z_{l}=-1\pm \varepsilon \sum_{n=1}^{\infty }a_{n}\sin(2n-1)\lambda x.$$
where $a_{n}=\pm \frac{2}{\pi (2n-1)}$.

Corresponding spectral densities and the corrugation function *ξ*_4_ are:(77)$$\Omega_{11}(\lambda )=\Omega_{22}(\lambda )=\Omega_{12}(\lambda )=\frac{1}{4}\sum_{n=1}^{\infty }a_{n}^{2}[\delta (\omega -n\lambda )+\delta (\omega +n\lambda )],$$
(78)$$\xi_{4}=\sum_{n=1}^{\infty }a_{n}{}^{2}\frac{b_{jk}(n\lambda ,Ha)}{2A_{1}}=\sum_{n=1}^{\infty }\frac{4b_{jk}(n\lambda ,Ha)}{2\pi^{2}{\left(2n-1\right)}^{2}A_{1}}.$$
where,
(79a)$$k_{1n}=\sqrt{\frac{2n^{2}\lambda^{2}+Ha^{2}+Ha\sqrt{Ha^{2}+4n^{2}\lambda^{2}}}{2}},$$
(79b)$$k_{2n}=\sqrt{\frac{2n^{2}\lambda^{2}+Ha^{2}-Ha\sqrt{Ha^{2}+4n^{2}\lambda^{2}}}{2}}.$$

## 4. Results and Discussion

Taking the same parameter values, through comparison, we found that our present result is in exact agreement with the findings of Buren and Jian [34]. However, compared with those [34], we extended the shape of the wall from sine to random shape, and the calculation process was simplified. The influence of *Ha* on the corrugation function *ξ* for different values of wave number *λ* is illustrated in Figure 5, when the corrugations are in the phase *θ* = 0. Increases in Ha always reduce the impact of corrugations on the EMHD flow for given *λ*. When *Ha* is fixed, it was found that the larger *λ* leads to larger roughness function, which means more driving force is needed to actuate the flow. Within the low range 0 < *Ha* < 2, the corrugation function declines sharply, and then becomes gentle gradually for a higher *Ha* number. In addition, it can be seen from Figure 5 that, when wave number is small (such as *λ* = 0.4), the wall roughness has almost no effect on the flow.

In order to generalize the result of Buren and Jian [34], we used the same parametrical values to study the triangular shape of the wall roughness in Figure 6. Figure 6a shows that, when *λ* is fixed, the roughness function decreases with the *Ha* number. When 0.1 < *λ* < 0.2, the roughness function has a little discrepancy for different *Ha*. However, when *λ* > 0.2, *ξ* increases with *λ*, and a lower value of *Ha* results in larger roughness function. Figure 6b shows that, with the increase of *Ha*, the roughness function *ξ* constantly decreases and gradually attains a stable status.

The reason for this is that the electromagnetic force can suppress the effect of roughness and reduce the loss caused by the resistance, which is related to the second-order problem.

Figure 7 and Figure 8 show the variations of the corrugation function *ξ* with *Ha* and *λ* for rectangular and sawtooth shape roughness, respectively. the corrugation function is always positive, which shows that the flow rate always reduces with the increase of roughness when the average pressure drop and electromagnetic fields are prescribed.

Similarly, we can study the effect of phase difference between upper and lower plates on EMHD flow. For our present four kinds of roughness, only sinusoidal corrugation strictly has a conception of phase difference, and this problem has been studied by Buren and Jian [34]. For other kinds of roughness, we only consider two special cases of *θ* = 0 (in phase) and *θ* = π (out of phase).

The variations of triangular, rectangular, and sawtooth corrugation function with Ha and *λ* for phase difference *θ* = 0 (in phase) and *θ* = π (out of phase) are illustrated in Figure 9, Figure 10 and Figure 11. However, in the present derivation, we found that only the Fourier coefficients for different shapes of ripples are different, and these coefficients determine the magnitude of the corrugation function. However, the physical meaning behind Figure 10 and Figure 11 is the same as Figure 9, which reveals that the corrugation function in phase of two plates is smaller than that when the roughness is out of phase. Furthermore, it means that the flow resistance becomes small, and the flow rate increases for in-phase roughness compared with the one out of phase.

Form Figure 12, we can note that, although the three corrugations have the same trend, the roughness of rectangular is significantly greater than that of triangular roughness and sawtooth roughness. The reason for this difference, mathematically speaking, is that their Fourier coefficients are different. This inspired us to make the microchannel rough shape triangular when we want to increase flow, and rectangular when we want to promote mixing.

## 5. Conclusions

We have investigated the impact of small random transverse wall roughness on EMHD flow in a microchannel based on random function theory. The stream function is expanded with small parameters (amplitude of the random roughness), and some properties of stationary random process are employed to gain analytical expression of the random corrugation function *ξ*, which is a measure of flow rate deviated from the case with no roughness on the two plates. Using the method presented in this paper, we can study the corrugation function of random shapes. When the ripple is sinusoidal, we find the final expression of the corrugation function is the same as that obtained by Buren and Jian [34]. No matter what kind of corrugation, identical conclusions can be drawn that increasing wavenumber *λ* and decreasing *Ha* number result in a growth of the corrugation function *ξ*. These results mean that the electromagnetic force can suppress the effects of roughness, increase flow, and promote production. In addition, we studied the influences of phase differences *θ* = 0 and *θ* = *π* on the corrugation function for all kinds of roughness. We found that the flow resistance becomes small and the flow rate increases for in-phase roughness compared with the one out of phase. In the future, we can use this method to study electroosmosis flow in microchannel with random roughness.

## Figures and Tables

**Figure 1 micromachines-14-01617-f001:**
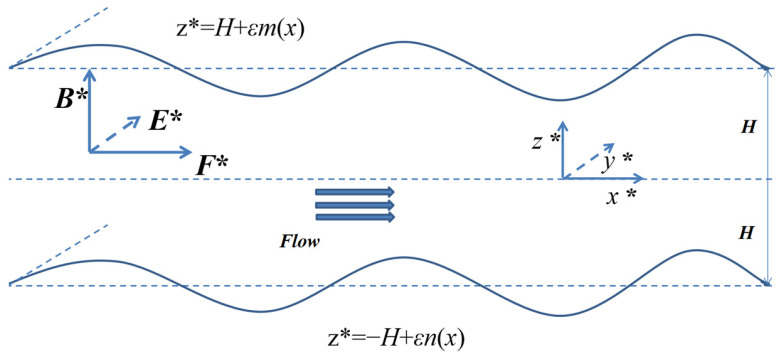
Schematic illustration of EMHD flow through a microchannel with a random roughness wall.

**Figure 2 micromachines-14-01617-f002:**
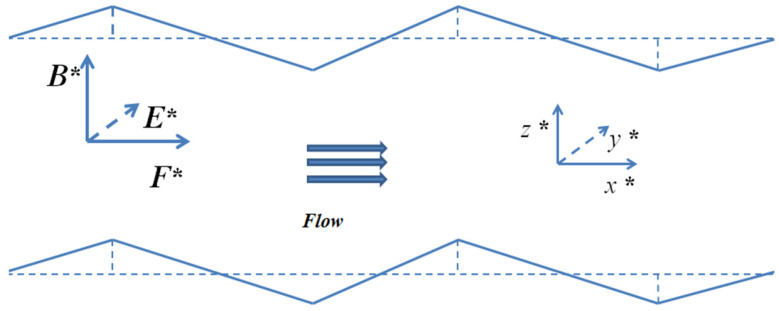
Schematic diagram of triangular ripple EMHD flow.

**Figure 3 micromachines-14-01617-f003:**
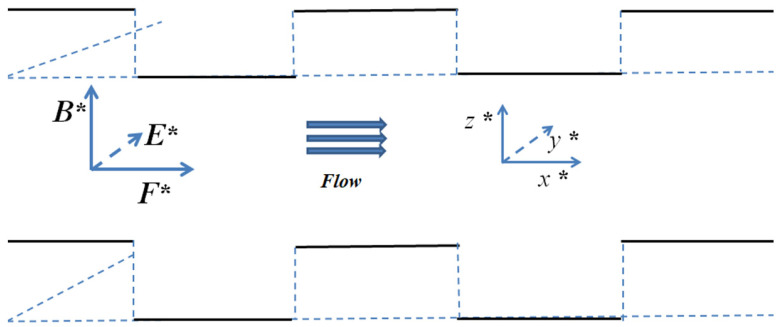
Schematic diagram of rectangular ripple EMHD flow.

**Figure 4 micromachines-14-01617-f004:**
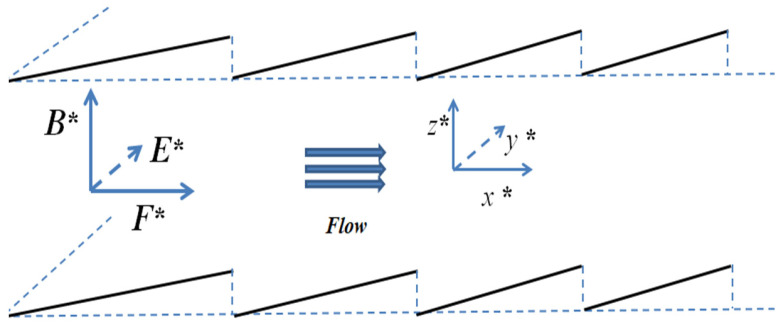
Schematic diagram of sawtooth ripple EMHD flow.

**Figure 5 micromachines-14-01617-f005:**
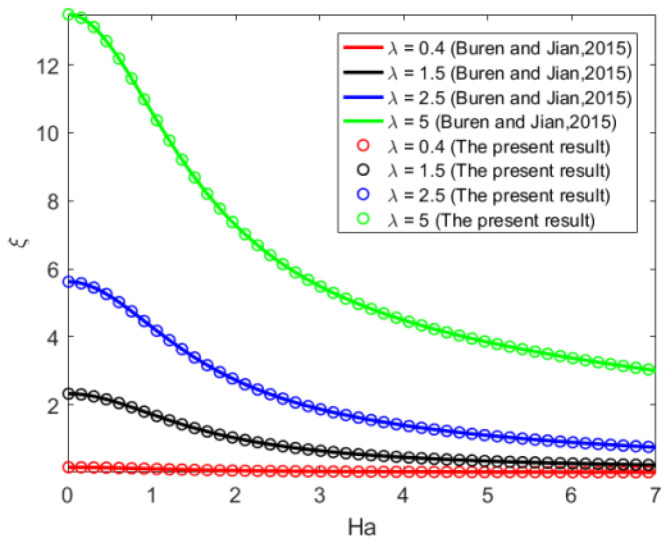
Sinusoidal corrugations variations of *ξ* with *Ha* for different *λ* (*θ* = 0) [34].

**Figure 6 micromachines-14-01617-f006:**
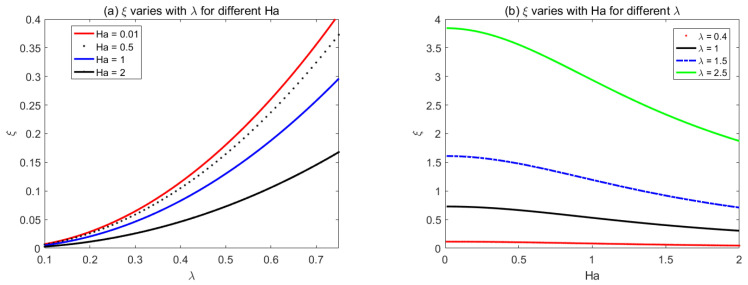
The variations of triangular corrugation function *ξ* with (**a**) *λ* for different *Ha*, (**b**) Ha for different *λ*.

**Figure 7 micromachines-14-01617-f007:**
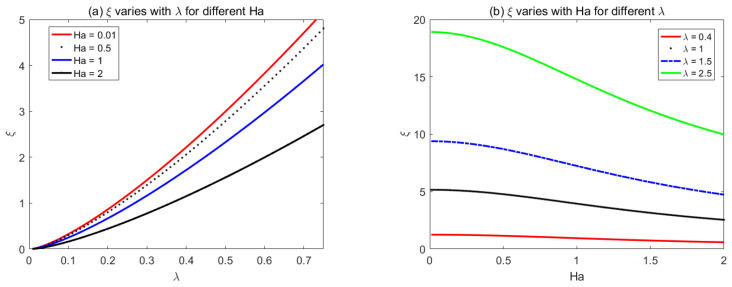
The variations of rectangular corrugation function *ξ* with (**a**) *λ* for different *Ha*, (**b**) *Ha* for different *λ*.

**Figure 8 micromachines-14-01617-f008:**
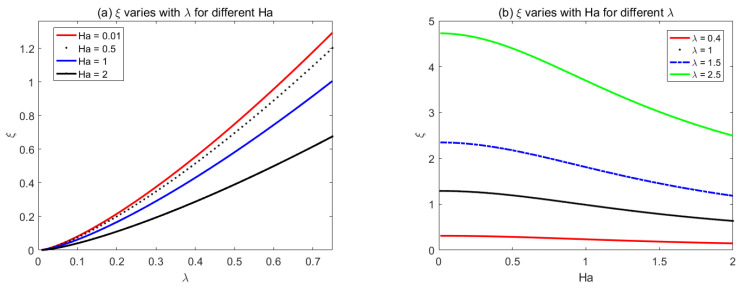
The variations of sawtooth corrugation function *ξ* with (**a**) *λ* for different *Ha*, (**b**) *Ha* for different *λ*.

**Figure 9 micromachines-14-01617-f009:**
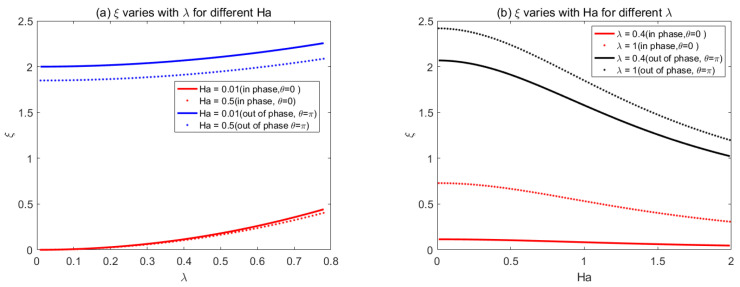
Variations of triangular corrugation function *ξ* with (**a**) *Ha* and (**b**) *λ* for phase difference *θ* = 0 (in phase) and *θ* = *π* (out of phase).

**Figure 10 micromachines-14-01617-f010:**
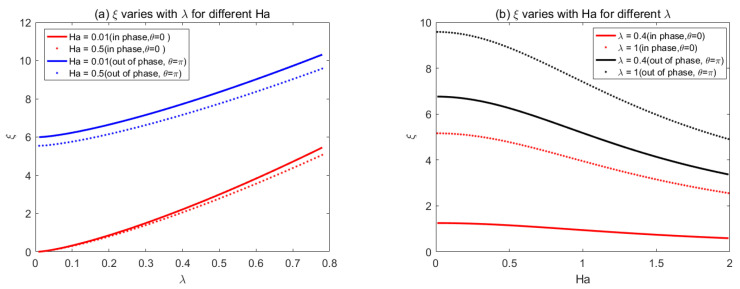
Variations of rectangular corrugation function *ξ* with (**a**) *Ha* and (**b**) *λ* for phase difference *θ* = 0 (in phase) and *θ* = *π* (out of phase).

**Figure 11 micromachines-14-01617-f011:**
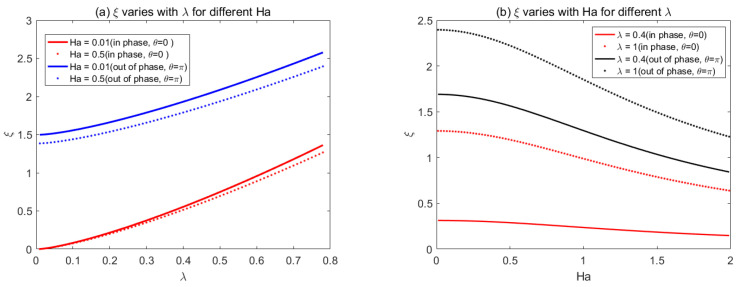
Variations of sawtooth corrugation function *ξ* with *Ha* and *λ* for phase difference. *Θ* = 0 (in phase) and *θ* = *π* (out of phase).

**Figure 12 micromachines-14-01617-f012:**
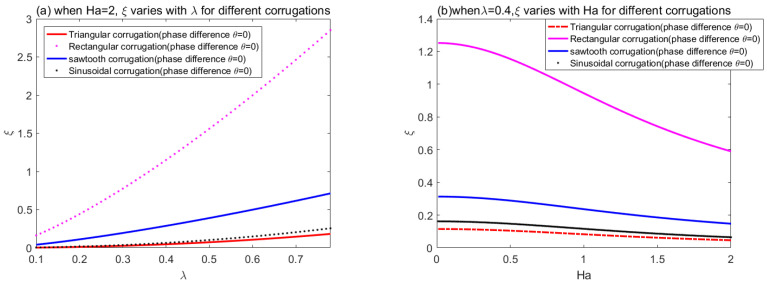
Variations of corrugation functions *ξ* (**a**) with *λ* and (**b**) Ha for different corrugations.

## Data Availability

Data sharing not applicable.

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
