# Peer review of "Electromagnetohydrodynamic (EMHD) Flow in a Microchannel with Random Surface Roughness"

_micromachines, 2023, doi:10.3390/mi14081617_

Round 1

Reviewer 1 Report

This work expanded the scope of surface roughness profile from sinusoidal (described in doi.org/10.1002/elps.201500029) to random for EMHD flow systems. It’s interesting that the final expression of corrugation function stays the same regardless of the surface roughness profile. The paper is helpful in rational design of EMHD based microfluidic systems and can be considered for publication after making the following minor edits:

1. Abstract: please clarify where the subject of study is one microchannel with two parallel plates or ‘two microchannels’ as mentioned in line 9. 

2. It would be very helpful to include a nomenclature list to guide readers through the work.

3. Line 299: this is the concept ‘in phase’ and ‘out of phase’ first introduced in the manuscript. It can be interpreted from the later results section that this is the periodic phase of the roughness function/profile. However, phase is also frequently referred to describe the liquid in fluidic systems. To avoid confusion for general audience, please provide another sentence or two to clarify the concept here.

4. The difference in levels of the roughness function should be pointed out/discussed for the four roughness profiles, e.g. rectangular corrugation function have significantly higher roughness function value than the other types. Please also discuss how these results tie to rational designs of EMHD based microfluidic systems.

5. Figure 9-11: please specify the difference between blue and red groups in (a) and between black to red groups in (b) with labels in the graph and context in the legends.

The manuscript is overall nicely written and has good quality of English language. Readers should be able to follow through without any questions. Minor edits could be made to further improve the fluency though. For example, the sentence in line91 can be adjusted to 'Section 5 provides a summary of the full text.'

Author Response

Thanks for your comments, we have made the following revisions to the manuscript. See the attachment for the modification.

Reviewer 2 Report

There are several grammatical mistakes. Please work close to a native English speaker to refine the language of this manuscript. There are significant concerns about the grammar, usage, and overall readability of the manuscript. Therefore, the request is to revise the text to fix the grammatical errors and improve the overall readability of the text before this work is considered for publication.

Author Response

(The authors gave the same response as above.)

Author Response

(The authors gave the same response as above.)

Round 2

Reviewer 2 Report

After checking through the revised version, it is worth mentioning that the authors have satisfactorily responded to all the questions and made the necessary changes to the manuscript. I have no further questions and suggest accepting the revised manuscript.